

# BVOC emissions from English oak (*Quercus robur*) and European beech (*Fagus sylvatica*) along a latitudinal gradient

Ylva Persson[1], Guy Schurgers[2], Riikka Rinnan[3] and Thomas Holst[1]

[1]Department of Physical Geography and Ecosystem Science, Lund University, Sölvegatan 12, 223 62 Lund, Sweden
[2]Department of Geosciences and Natural Resource Management, University of Copenhagen, Øster Voldgade 10, DK-1350 Copenhagen K, Denmark
[3]Terrestrial Ecology Section, Department of Biology, University of Copenhagen, Universitetsparken 15, 2100 Copenhagen E, Denmark

*Correspondence to*: Y. Persson, (ylva.persson@nateko.lu.se)

**Abstract.** English oak (*Quercus robur*) and European beech (*Fagus sylvatica*) are amongst the most common tree species growing in Europe, influencing the annual Biogenic Volatile Organic Compound (BVOC) budget in this region. Studies have shown great variability in the emissions from these tree species, originating from both genetic variability and differences in climatic conditions between study sites. In this study, we examine the emission patterns for English oak and European beech in genetically identical individuals and the potential variation within and between sites. Leaf scale BVOC emissions, net assimilation rates and stomatal conductance were measured at the International Phenological Garden sites of Ljubljana (Slovenia), Grafrath (Germany) and Taastrup (Denmark). Sampling was conducted during three campaigns between May and July 2014.

Our results show that English oak mainly emitted isoprene whilst European beech released monoterpenes. The relative contribution of the most emitted compounds from the two species remained stable across latitudes. The contribution of isoprene for English oak from Grafrath and Taastrup ranged between 92–97% of the total BVOC emissions, whilst sabinene and limonene for European beech ranged between 30.5–40.5% and 9–15% respectively for all three sites. The relative contribution of isoprene for English oak at Ljubljana was lower (78%) in comparison to the other sites, most likely caused by frost damage in early spring. The variability in total leaf-level emission rates from the same site was small, whereas there were greater differences between sites. These differences were probably caused by short-term weather events and plant stress. A difference in age did not seem to affect the emission patterns for the selected trees.

This study highlights the significance of within-genotypic variation of BVOC emission capacities for English oak and European beech, the influence of climatic variables such as temperature and light on emission intensities and the potential stability in relative compound contribution across a latitudinal gradient.



## 1 Introduction

Plants act as important contributors of gases and particles released into the atmosphere, which might alter the atmospheric composition (Atkinson, 2000; Skjøth et al., 2008). One group of molecules consisting of a large variety of sizes and properties and which plays numerous roles in both biotic and atmospheric processes is called Biogenic Volatile Organic

Compounds (BVOCs) (Guenther et al., 1995; Kesselmeier and Staudt., 1999; Karl et al., 2009; Laothawornkitkul et al., 2009; Peñuelas and Staudt, 2010). BVOCs have a biotic role as they are involved in pollinator attraction and plant defence against biotic and abiotic stresses (Langenheim, 1994; Kost and Heil, 2006; Maffei, 2010; Pierik et al., 2014). They also contribute to particle growth, formation of cloud condensation nuclei and OH reactivity (Atkinson, 2000; Di Carlo et al., 2004; Peñuelas and Staudt, 2010; Paasonen et al., 2013). Each plant species has a specific blend of BVOCs and this blend

can vary strongly between individuals, species and community structures (Skjøth et al., 2008; Peñuelas and Staudt, 2010; Bäck et al., 2012).

In this study, we address the emission of terpenes, which can be divided into isoprene (with a carbon skeleton consisting of a $C_5$ unit), monoterpenes (two $C_5$ units) and sesquiterpenes (three $C_5$ units). Apart from BVOC emissions induced by plant stress, emitted compounds are regarded as both light and temperature dependent, or being dependent on temperature only.

The dependency is largely determined by the extent of released *de novo* emissions (Kesselmeier and Staudt, 1999; Dindorf et al., 2006). A *de novo* emitter is directly dependent on net photosynthetic rates and gets its carbon for BVOC production from recently synthesized photosynthesis intermediates. Stored BVOCs can exist in various storage structures and are therefore independent of on-going primary production (Lerdau et al., 1997; Šimpraga et al., 2011). The temperature and light dependencies, together with photosynthetic rates and regional differences in vegetation composition cause variations in

space and time in BVOC emissions (Guenther et al., 1995; Kesselmeier and Staudt., 1999; Sharkey and Yeh, 2001; Skjøth et al., 2008).

For many tree species, high BVOC production in comparison to other plant taxa is related with plant size and their long life span (Holopainen, 2011). In Europe, some of the most common broadleaved tree species are English oak (*Quercus robur*) and European beech (*Fagus sylvatica*) (Simpson et al., 1999; Skjøth et al., 2008). English oak is considered to be a strong

isoprene emitter (Isidorov et al., 1985; Kesselmeier and Staudt, 1999; Pokorska et al., 2012; Steinbrecher et al., 2013) and European beech is a moderate or strong monoterpene emitter (Tollsten and Müller, 1996; Kesselmeier and Staudt, 1999; Dindorf et al., 2006; Holzke et al., 2006; Kleist et al., 2012). Both trees are reported to be *de novo* emitters (Holzke et al., 2006; Kleist et al., 2012; Steinbrecher et al., 2013). Isoprene has a high Henry's law constant and therefore partitions mainly to the gas phase, and European beech lacks specialized storage structures for the monoterpenes produced (Niinemets et al.,

2004; Demarcke et al., 2010; Kleist et al., 2012). European beech is most abundant in central Europe, has a northern limit in Southern Sweden, Poland and Ukraine and a southern limit in Spain and Portugal. English oak has a northern growing limit in Scandinavia, Russia and the Baltic countries, but it has no southern limit in Europe (Skjøth et al., 2008). The spatial





variability of these two trees means the climatic conditions they experience have a large range, but how this range of climates affects the trees' BVOC emission patterns is relatively unknown (Simpson et al., 1999; Skjøth et al., 2008).

Modelling studies quantifying BVOC emissions for Europe typically combine either inventories or modelled distributions of tree species with a standardized emission rate for each species, which is adjusted temperature- and light-dependently. The simulated emission rates are often assumed to be constant over wide areas and either scaled up or down based on prevailing meteorological conditions (Simpson et al., 1999; Skjøth et al., 2008; Schurgers et al., 2011; Oderbolz et al., 2013). To improve models with regard to the handling of BVOC emission variability on regional or global scales, there is a need to separate the climatic impact from genetic variation. In a study by Funk et al. (2005) performed on 60 Red Oak (*Quercus rubra*) trees grown both indoors and outdoors, there was a twofold difference in isoprene emission at standardized light and temperature conditions amongst individuals. Staudt et al. (2001) measured 146 individuals of Holm oak (*Quercus ilex*) and concluded that the individuals could be divided into three main chemotypes with respect to the proportion of the major compounds emitted. This division was fairly independent in relation to season, leaf age and emission rates. Bäck et al. (2012) also divided 40 Scots pine (*Pinus sylvestris*) trees into emission chemotypes, which remained fairly stable over time. The conclusion for all three studies was that the emission patterns were governed by genetic variation rather than temperature and light, but the influence of climatic variables could not be excluded (Staudt et al., 2001; Funk et al., 2005; Bäck et al., 2012).

In this study, leaf scale BVOC emission patterns of English oak (*Quercus robur*) and European beech (*Fagus sylvatica*) were investigated in genetically identical individuals grown under natural conditions at three sites across Europe. The aim of this study was to (i) evaluate if the emission patterns differed within individuals and between growing locations and (ii) to get a better understanding of the governing factors behind observed BVOC variations. By conducting measurements on genetically identical individuals, we could exclude genotypic variations in the emission characteristics and focus on the emission patterns induced by climatic variation.

## 2 Methods

### 2.1 Site description and weather conditions

Three measurement campaigns were carried out from May to July 2014 at three International Phenological Garden (IPG) sites in Europe. The IPG network was initiated in 1957 and performs long-term phenological observations on some of the most common European plant species across Europe. All trees used in the network are genetically identical clones, which mean the genetic variation between sites is absent (Chmielewski et al., 2013). The visited sites followed a latitudinal transect consisting of IPG 055 in Ljubljana, Slovenia (46°04´ N, 14°30´ E), IPG 036 in Grafrath, Germany (48°18´ N, 11°17´ E), and IPG 010 in Taastrup, Denmark (55°40´ N, 12°18´ E). For the distribution of European beech, the sites are all within their natural distribution ranges, whilst only Taastrup is within the range for English oak (Fig. 1). Measurements were performed





from South to North with campaigns of approximately two weeks each in order to minimize differences in leaf developmental stages between sites.

The long-term average air temperature and precipitation experienced on site generally decreased with increasing latitude (Table 1). The weather conditions for the visited sites differed during sampling (8:00–16:00), where the daily precipitation

and average temperature during the campaigns and two weeks prior to measurements can be found in Fig. 2. In Ljubljana, unstable weather conditions led to reoccurring thunderstorms and average temperatures below 20 °C during the whole measurement campaign. In the beginning of the campaign in Grafrath, there were a couple of days with an average daily temperature between 12.4–14.3 °C before it rose to an average of 21.2–24.5 °C. Clear, sunny conditions were experienced during the whole campaign with the exception of two short thunderstorms in the beginning of the measurement period. The

highest temperature experienced reached up to 38.7 °C. Taastrup had one day of intense rainfall of 75 mm in the beginning of the campaign. The rest of the campaign experienced cloudy but calm weather with temperatures around 17.2–22.7 °C.

The leaf unfolding date for English oak and European beech was typically between 23–27 April for Ljubljana and Grafrath, whilst European beech in Taastrup unfolded its leaves 5 May. There are no long-term observational data for the leaf unfolding of English oak in Taastrup. In 2014, the leaf unfolding started 12–21 days earlier on all sites in comparison to the

long-term average for all trees. The exception was for the European beech in Grafrath, which followed the average time of unfolding (Table 2).

## 2.2 BVOC sampling and analysis, net assimilation and stomatal conductance

Measurements were made on five English oak trees (one in Ljubljana, two in Grafrath and two in Taastrup) and four European beech trees (one in Ljubljana, two in Grafrath and one in Taastrup). In Grafrath, only five year old English oaks

were available, whilst the trees at the other sites were 42–43 years old. For European beech, all trees were between 43 and 51 years old. Measurements were made on fully developed leaves and one sample was taken per leaf. For English oak, 5–9 leaves were measured per tree, giving a total of 36 samples. For European beech, 7–17 leaves were measured per tree, which gives a total of 49 samples. All samples were taken on the lowest situated branches of the tree (1–2 m above ground). This height was chosen as earlier results in Taastrup had shown little emission pattern difference for English oak and different

heights within the tree due to the wide spacing between trees (Persson et al., 2016). Because there existed a similar wide spacing between trees in Ljubljana and Grafrath, sampling the lowest branches was considered appropriate.

In addition to BVOC measurements, net assimilation rates (A) and stomatal conductance ($g_s$) were determined for each leaf using a portable photosynthesis system (LI-6400, LICOR, Lincoln, NE, USA), equipped with a LED source leaf chamber (6400-02B). The system allows control over $CO_2$, light, temperature and humidity within the chamber, and the

measurements were made under fixed environmental conditions. The leaf within the chamber was acclimated to 1000 µmol $m^{-2}$ $s^{-1}$ Photosynthetic Active Radiation (PAR), 400 µmol $CO_2$ mol air$^{-1}$, and 50–65% relative humidity before BVOC sampling. An acclimation period of one hour was considered to be sufficient to reach a constant release of BVOCs. The inlet





air stream (700 ml min$^{-1}$) entering the IRGA was filtered through a hydrocarbon trap in order to remove organic contaminants from the sample air stream. Leaf temperatures were held constant at the anticipated average daily temperature (20–30°C depending on the site). A 5–L air sample for BVOC analysis was taken directly from the chamber outlet and passed through a stainless steel cartridge (Markes International Limited, Pontyclun, UK) packed with adsorbents Tenax TA

(porous organic polymer) and Carbograph 1TD (graphitized carbon black). The air was extracted from the leaf chamber with the help of flow controlled pocket pumps (Pocket Pump, SKC Ltd., Dorset, UK) with a sampling flow rate of 200 ml min$^{-1}$. Empty chamber blanks were also collected under the same chamber conditions in order to acknowledge and remove any background contamination. After the air samples were collected, an assimilation and intercellular response curve (A-C$_i$) was run on every second leaf. The intercellular $CO_2$ concentration (Ci) used in the A-C$_i$ curve is derived by the gas exchange

calculations implemented into the LI-6400 system (for further details on the equations used by the instrument, see von Caemmerer and Farquahar, 1981). In order to get the A-C$_i$ response, the $CO_2$ level was changed within the chamber. The $CO_2$ amount was first decreased from 400 to 100 µmol mol$^{-1}$ in 100 µmol mol$^{-1}$ steps and a further 50 µmol mol$^{-1}$ to reach the lowest level. The $CO_2$ level was then increased to 400 µmol mol$^{-1}$, where two points were logged in a row to allow the leaf some recovery. Lastly, the $CO_2$ level was increased to 1000 in 200 µmol mol$^{-1}$ steps. This resulted in 10 logged points.

For each step, there was an adjustment time of at least 90 seconds in order to reach stability. If stability was not reached after 150 seconds, logging would occur anyway. The water use efficiency (WUE) for the leaves was calculated by dividing the net assimilation rate with the transpiration rate. Lastly, the dry weight of the leaves was determined.

The sample cartridges were sealed with Teflon coated brass caps immediately after sampling, and stored in a refrigerator at 2°C and analysed within two weeks. The samples were analysed using a gas chromatograph-mass spectrometer (7890A

Series GC coupled with a 5975C inert MSD/DS Performance Turbo EI System, Agilent Santa Clara, CA, USA) after being thermally desorbed (UNITY2 coupled with an ULTRA autosampler, Markes, Llantrisant, UK). Helium was used as the carrier gas whilst the oven temperature was held at 40 °C for 1 min, raised to 210 °C at a rate of 5 °C min$^{-1}$ and further up to 250 °C at a rate of 20 °C min$^{-1}$. The BVOCs were separated using a HP-5 capillary column (50 m, diameter 0.2 mm and film thickness 0.33 µm). The compounds were identified according to pure standard solutions and the mass spectra in the Wiley

data library (Wiley and Sons Ltd., Chichester, UK). Standard solutions were injected into adsorbent cartridges in a stream of helium and analysed as samples. The standards available were isoprene, α-pinene, camphene, δ-phellandrene, limonene, eucalyptol, γ-terpinene, linalool, aromadendrene and α-humulene in methanol (Fluka, Buchs, Switzerland). To quantify BVOCs to which no standards were available, α-pinene was used for MTs and α-humulene was used for SQTs. All chromatograms were analysed with the MSD ChemStation Data Analysis software (G1701CA C.00.00 21 Dec 1999; Agilent

Technologies, Santa Clara, CA, USA).



## 2.3 BVOC standardization

Since previous studies have shown that both English oak and European beech have light and temperature dependent emissions (Niinemets et al., 2004; Dindorf et al., 2006; Demarcke et al., 2010; Kleist et al., 2012), all emission rates were standardized according to the temperature and light dependent model of Guenther et al. (1993, 1995). The Eq. (1) is as follows

$$I = I_S \cdot C_L \cdot C_T, \text{ where } C_L = \frac{\alpha C_{L1} PAR}{\sqrt{1+\alpha^2 PAR^2}} \text{ and } C_T = \frac{\exp\frac{C_{T1}(T-T_S)}{RT_S T}}{1+\exp\frac{C_{T2}(T-T_M)}{RT_S T}} \quad (1)$$

where I stands for isoprene emission rate at temperature T (K) and PAR flux (µmol m$^{-2}$ s$^{-1}$) and $I_S$ is the isoprene emission rate at a standard temperature $T_S$ (303 K) and a standard PAR flux (1000 µmol m$^{-2}$ s$^{-1}$). $C_L$ is a factor for light dependence of the compound, where α= 0.0027 and $C_{L1}$= 1.066 are empirical coefficients. $C_T$ is a factor for temperature dependency, where R is the universal gas constant (8.314 J K$^{-1}$) and $C_{T1}$= 95 000 J mol$^{-1}$, $C_{T2}$= 230 000 J mol$^{-1}$ and $T_M$= 314 K are empirical coefficients. The standardization was done in order to be able to compare samples which had experienced different daily temperatures. In our measurements, the same PAR conditions of 1000 µmol m$^{-2}$ s$^{-1}$ were used on all samples, resulting in $C_{L1}$= 1. Temperature varied between 293–304 K and $I_S$ was determined by combining the observed I and the computed $C_T$.

## 2.4 Statistical analysis

A one way ANOVA followed by Tukey's test was performed to test if the total BVOC emissions differed between sites and a paired t-test was made to test for similarities between clones from the same site.

Principal component analysis (PCA) was used in order to investigate whether the individual trees differed from each other within and between sites according to their individual BVOC emission patterns. The PCA was run through SIMCA (Umetrics, version 13.0.3.0, Umeå, Sweden) after centering and unit variance-scaling of the variables. Two outliers from English oak with extremely high emissions were removed from the dataset due to suspected rough handling of the leaves.

## 3 Results

### 3.1 Photosynthetic rates and A-C$_i$ responses

The net assimilation rate at standard $CO_2$ conditions (400 µmol mol$^{-1}$) for English oak ranged between 1.58 and 7.00 µmol $CO_2$ m$^{-2}$ s$^{-1}$, with the lowest rates found in Ljubljana and one of the Grafrath trees and the highest rate found in the other Grafrath tree. Similarly, $g_s$ ranged between 0.01 and 0.07 mol $H_2O$ m$^{-2}$ s$^{-1}$, with the lowest rates found in Ljubljana and the highest in the second tree growing in Grafrath. The WUE had a range of 2.78–8.36 µmol mol$^{-1}$ (Fig. 3).





For European beech, the average net assimilation rate ranged between 2.56 and 8.93 µmol $CO_2$ m$^{-2}$ s$^{-1}$, with the lowest rates in Taastrup and the highest rate in Ljubljana (Table 3). $g_s$ was between 0.03 and 0.14 mol $H_2O$ m$^{-2}$ s$^{-1}$. The WUE did not vary considerably between sites, with a range of 5.29–6.00 µmol mol$^{-1}$ (Fig. 4).

In total, 3–4 A-$C_i$ curves were taken per tree for English oak and 3–10 for European beech. The A-$C_i$ curves for English oak showed a similar pattern as with the net assimilation rate and $g_s$, with less response from Ljubljana and one of the trees in Grafrath and a higher response for the other tree in Grafrath and in Taastrup. There was a clear difference in the $CO_2$ response between the trees in Grafrath, but no clear difference from the trees growing in Taastrup (Fig. 5). All the European beech trees showed a similar response with increasing $CO_2$ levels, where the net assimilation rate tended to level off at 800 µmol $CO_2$ mol$^{-1}$. Taastrup has the lowest $CO_2$ response in regards to different $CO_2$ levels, but without distinct differences from the other trees (Fig. 6).

### 3.2 BVOC emission from English oak and European beech

The ranges of the total standardized BVOC emission rates from the English oak and European beech trees differed considerably between sites (Fig. 7). The English oak in Ljubljana had low emission with no high values, and the emission was lower in comparison to the other sites (Fig. 7a; $P<0.001$ for Tukey's test). The highest emission variation was found for tree number two in Grafrath. One of the European beech trees in Grafrath had the highest emission variation, whilst the tree in Taastrup had the lowest variation and significantly lower emissions than the other sites (Fig. 7b; $P<0.01$). For both English oak and European beech, the total BVOC emissions from the cloned trees from the same site were not significantly different from each other ($P>0.05$).

The total average standardized BVOC emission rates for all samples from English oak ranged between 0.25 and 3.35 µg gdw$^{-1}$ h$^{-1}$, with the highest emissions in Grafrath and the lowest emissions in Ljubljana. The largest contribution to the overall emission came from isoprene, which made up 78–97% of the total BVOC emission (Table 3). Each sample contained between 3–6 detectable compounds which, apart from isoprene, were α-pinene, camphene, 3-carene, limonene and β-ocimene.

The mature oak in Ljubljana had lowest emissions, ranging from 0.05–0.79 µg gdw$^{-1}$ h$^{-1}$ between samples. From the two five year old trees in Grafrath, the emission rate ranged between 1.22–3.76 µg gdw$^{-1}$ h$^{-1}$ for the first oak and between 0.79–6.1 µg gdw$^{-1}$ h$^{-1}$ for the second oak. In Taastrup, the emission rate ranged from 1.42–3.45 µg gdw$^{-1}$ h$^{-1}$ for the first mature oak and between 0.5–3.78 µg gdw$^{-1}$ h$^{-1}$ for the second oak.

The European beech had an average emission rate between 0.29–0.69 µg gdw$^{-1}$ h$^{-1}$, with the highest emission found at the site in Grafrath and the lowest emission in Taastrup. The total emission rate was fairly similar in both Ljubljana and Grafrath, with a range of emission of 0.41–1.03 µg gdw$^{-1}$ h$^{-1}$ in Ljubljana and 0.09–1.71 µg gdw$^{-1}$ h$^{-1}$ in Grafrath. Taastrup had the lowest standard emission rate, ranging 0.19–0.41 µg gdw$^{-1}$ h$^{-1}$ (Table 3). Each sample contained between 2–10 detectable compounds. In addition to sabinene, which had the highest compound contribution, these were α-thujene, α-pinene, camphene, sabinene, α-phellandrene, 3-carene, α-terpinene, limonene, β-phellandrene, γ-terpinene and terpinolene.



### 3.3 Compound variation across latitudes

Both the individual samples and the relative emission contribution of major emitted compounds varied between sites for both English oak and European beech (Figs 8 and 9). For English oak, the percentage of compound contribution from isoprene varied between 91–97% apart from the tree growing in Ljubljana where the contribution was 78%. In Ljubljana there was a higher variation between samples in comparison to the other sites, where a few samples had high isoprene emission whilst the remaining samples contained more monoterpenes. Other major emitted compounds were limonene, 3-carene and α-pinene (in decreasing order), which was a consistent pattern across sites (Fig. 8). For European beech, sabinene was the main emitted compound with a contribution 30.5–40.5%. Another important emitted compound was limonene, which contributed between 9–15%. In the few samples where sabinene had not been detected, limonene was the main emitted compound. The remaining compounds varied in importance depending on the site. γ-terpinene was an important compound in Ljubljana and Grafrath, but not emitted at all in Taastrup. Taastrup also had the highest relative contribution of sabinene within samples, when it was emitted, and less compound variation in comparison to the other cloned trees (Fig. 9).

PCA was used to describe latitudinal differences based upon the BVOC emission profiles (Figs 10 and 11). The English oak trees showed a clear separation between sites, but less within (Fig. 10). The first PC explained 60.9% of the variation in the emission of individual BVOCs and it separated the samples from Grafrath from Ljubljana and Taastrup. It also described the difference in emission profiles with regard to the average daily temperature, with the highest temperatures during sampling where found with high values on PC1. The second PC explained 18.0% of the variation and separated the samples from Ljubljana and Taastrup. The emission profiles within a site were not separated from each other. The average daily PAR did not correlate with the emission profiles. The loading plot revealed that the Grafrath samples were characterized by higher relative emissions of other compounds than the monoterpene β-ocimene, which was a characteristic compound for Taastrup (Fig. 10).

Figure 11 shows the PCA score plot and loading plot for European beech. Here, the separation between sites based on the beech BVOC emission profiles was less distinct in comparison to English oak. There was no clear separation or clustering of the samples in relation to PAR or temperature. The first PC explained 50.8% of the variation in the data and the second PC explained 17.2%. Taastrup, which had much lower PAR level during sampling than other sites, had lower relative emissions of all compounds except camphene, which was more characteristic for the site (Fig. 11).

### 4 Discussion

The amount emitted and the mixture of BVOCs reported in the literature can vary immensely for individual species (Kesselmeier and Staudt, 1999; Owen et al., 2001; Noe et al., 2012; Pokorska et al., 2012; Steinbrecher et al., 2013). Some studies suggest that this within-species variability is mostly due to genetic variation, but the majority of studies performed have either done measurements on mature and genetically different trees or on genetically identical but young and potted individuals (Lehning et al., 1999; Staudt et al., 2001; Funk et al., 2005; Bäck et al., 2012). One of the aims of this campaign



was to study if similar emission patterns were observed for genetically identical individuals in order to further confirm this genetic dependency. By using the IPG network, where common European tree species are cloned and dispersed across Europe, it is possible to measure the emission pattern variability under natural field conditions for the same genotype grown at different locations. And as the network has been in practice since 1957 (Chmielewski et al., 2013), many of the sites could

provide mature trees, which often have different BVOC emission rates and patterns in comparison to younger trees (Street et al., 1997; Csiky and Seufert, 1999; Thoss et al., 2007).

The IPG network represents a unique opportunity to examine genetically identical adult trees growing in different environments. This study builds on the work of a previous study done in Taastrup 2013, which concluded that there was little emission pattern difference between clones within the same species (Persson et al., 2016), and expands it towards a

latitudinal gradient using the same methods. The three campaigns presented here, showed that cloned trees from the same site did not provide significantly different emission profiles. We also showed that the main compounds emitted from English oak and European beech were the same at different sites, although there were some differences in the relative contribution. The most dominant compounds released were isoprene for English oak and sabinene for European beech which is in line with previous studies (Isidorov et al., 1985; Tollsten and Müller, 1996; Kesselmeier and Staudt., 1999; Dindorf et al., 2006;

Holzke et al., 2006; Kleist et al., 2012; Pokorska et al., 2012; Steinbrecher et al., 2013; Persson et al., 2016).

### 4.1 BVOC emission in regards to local growing conditions

The average standardized emission for English oak was between 0.25–3.35 µg gdw$^{-1}$ h$^{-1}$, which is a low estimate compared with other studies (Isidorov et al., 1985; König et al., 1995; Kesselmeier and Staudt, 1999; Pokorska et al., 2012 and references within; Persson et al., 2016). The lowest emission came from Ljubljana, which was expected to have the highest

BVOC emission rate due to its southern location. However, the oak tree was severely damaged by a frost event in the beginning of the year prior to the measurements, and a third of the tree had to be cut in order to save it (Ana Žust, personal communication). Both the frost and the cutting would have caused stress to the tree, and it is most likely that most resources would be allocated to recovery rather than BVOC emissions. A low net assimilation rate of 2.5 µmol $CO_2$ m$^{-2}$ s$^{-1}$ also indicated stress, which was less than half the rates found at the other sites, apart from one of the Grafrath trees. Furthermore,

$g_s$, WUE and the A-$C_i$ response were also low during the time of sampling, indicating closed stomata and low photosynthetic activity within the leaves. A similar pattern was shown for one of the young trees growing in Grafrath, which is also reflected in the lower BVOC emission capacity compared with the other clone growing at the same site (Figs 3 and 5). At Grafrath, there were only young English oak trees available for performing measurements as the only fully grown tree had died a couple of years earlier (Martin Piepenburg, personal communication). As the emission rates may vary with the age of

the tree (Street et al., 1997; Thoss et al., 2007), the age difference should be taken into account when the total emission rates are compared between sites. However, as the relative emission spectra remained stable between sites, it is likely when compared to the study made by Street et al. (1997) that an adult tree from Grafrath would emit a similar compound spectrum in comparison to the trees measured in this study.




The average standardized emission for European beech ranged between 0.29–0.69 µg gdw$^{-1}$ h$^{-1}$, which is low compared with other studies (Tollsten and Müller, 1996; Moukhtar et al., 2005; Holzke et al., 2006 and references therein). One reason for the low emission rates in this study could be that all the measurements were performed at the bottom part of the canopy. The beech in Taastrup has been shown to have a clear vertical emission profile, where the top of the canopy emitted 7–9 times

more in comparison to lower levels of the tree (Persson et al., 2016).

Another reason for the low emission rates could be the effect of other climatic influences than temperature and PAR. Beech is particularly drought sensitive and prone to adaptation depending on the local climatic conditions (Peuke et al., 2002). In this study, the average net assimilation rate for Ljubljana and Grafrath were between 4.62–8.93 µmol CO$_2$ m$^{-2}$ s$^{-1}$, whilst Taastrup had the lowest rate of 2.56 µmol CO$_2$ m$^{-2}$ s$^{-1}$. In comparison to Ljubljana and Grafrath, the leaves in Taastrup were

yellower in colour which might suggest some type of stress within the tree. Even though the net assimilation rate and g$_s$ were lower in Taastrup in comparison to the other sites, there was no clear difference in their WUE. Both the net assimilation rate and the transpiration rate followed a similar increasing or decreasing pattern, resulting in fairly stable WUE levels. This might be due to an ecotype adaptation suggested by Peuke et al. (2002), but longer and more detailed studies need to be conducted to confirm this hypothesis.

Even though studies have suggested that genetic variation has a strong influence on the BVOC emission patterns (Staudt et al., 2001; Funk et al., 2005; Bäck et al., 2012), local meteorological conditions are also expected to affect emission characteristics. It is well known that light and temperature have an important impact on plant BVOC emissions (see review by Grote et al. 2013; Li and Sharkey 2013), but other factors such as soil moisture, nutrient availability and biotic influences (e. g. herbivory) can further influence the emission patterns (Kesselmeier and Staudt, 1999; Peñuelas and Llusià, 2001;

Possell et al., 2004; Wu et al., 2015).

Past weather events have also been shown to alter the emission rates (Sharkey et al., 1999; Ekberg et al., 2009; Demarcke et al., 2010) and lead to different adaptation traits (Peuke et al., 2002). Sharkey et al. (1999) measured on Red oak and White oak and found that 95% of the basal emission rate of isoprene could be explained by the average level of temperature and PAR experienced over the last two days. Funk et al. (2005) used similar methods and found no clear relationship between

BVOC emissions and either past light or temperature, but suggested that this relationship could have been masked by a large intraspecific variation. Our findings demonstrate there is a separation of samples with regards to sites from English oak and with a slight connection to temperature, but it cannot be clearly attributed to the weather conditions only. The same goes for European beech, which showed even less clear distinctions between samples and sites (see Figs 10 and 11). This would suggest a more complex interaction between weather and the local growing conditions, which was not be assessed in this

study. Therefore, more studies are needed with more cloned individual trees and over different growing seasons in order to confirm this weather dependency.





## 4.2 Uncertainties

In this work, all measurements were conducted on the lowest positioned branches, as it was assumed to have little effect on English oak due to the wide spacing between trees. In Taastrup it was concluded that the wide spacing between trees caused similar light conditions at different height levels, and therefore resulted in similar emission patterns (Persson et al., 2016).

Other studies performed in denser forests have concluded that higher emissions are generally found at sunlit leaves (Bertin et al., 1997; Harley et al., 1997; Šimpraga et al., 2013). Furthermore, for Grafrath which only had five year old trees with an approximate height of 1 m, the difference between sunlit and shaded leaves was limited. However, as the trees in Ljubljana and Grafrath were standing with a comparable spacing between trees as the trees in Taastrup, we assume a similarly small difference between light intensities as were found in Taastrup in 2013. For European beech, it is more likely that there would have been an emission difference between the upper and lower part of the canopy. This needs to be taken into consideration when regarding these figures. But as all the trees have been measured at the approximately the same height, it is still possible to make comparisons between visited sites.

BVOC emission rates and spectra tend to differ over time. Holzke et al. (2006) measured a fully matured European beech tree over two vegetation periods to study the BVOC emission difference in time. Even though the tree experienced higher mean temperatures and longer warm periods, the emission rates measured in the latter year were lower than the former. It was explained to be caused by preceding weather events and in particular long periods without rainfall, which might have restricted leaf development. Leaf unfolding for European beech was approximately three weeks earlier in 2014 in comparison to 2013, but for both campaigns the beech leaves had at least 80 days to mature before measurements were taken. There are no leaf unfolding data for 2013 for English oak. When average temperatures and total amount of rainfall were compared, 2014 was a slightly wetter (+173 mm) and warmer (+2°C) year in comparison to 2013 seen over the whole year. During the period between May to July where measurements were taken, the weather did not differ considerably between 2013 and 2014 (data not shown). But when the emission rates from this study was compared with the emission rates in Taastrup (for more information, see Persson et al., 2016), the emission rates in 2013 were ten times higher for both English oak and European beech in comparison to 2014. In this case, past weather events do not explain the huge variation in BVOC emissions between the two measured years. However, the composition of compounds remained fairly stable, which could be an important aspect when it comes to modelling. This stability in the emission spectra has been seen for other trees as well (Staudt et al., 2001; Bäck et al., 2012), but the stability has usually only been tested over a few years. In order to prove that emission spectra remain stable over time, more studies over longer periods of time need to be done.

## 5 Conclusions

This work both highlights the potential stability in BVOC emission spectra across regions and discusses the climatic impact on English oak and European beech on trees that lack genetic variation. The IPG network, which studies the phenological patterns from genetically identical European tree species, has been used in order to better understand intraspecific variation





between sites and the causes behind them. The work further emphasizes the need to perform studies over longer timescales in order to fully understand the emission pattern variation.

Despite differences in age and location, the relative contributions of the highest emitted compounds were stable within as well as between studied sites in the latitudinal transect. However, the amounts of BVOC released differed between sites, even when standardized (Guenther et al. 1993, 1995) to the same temperature and light levels. This is most likely caused by past weather events and stress such as frost damage and tree cutting. The BVOC emission measured for English oak and European beech was lower than previous measurements by other studies, but the lower BVOC emission amounts were consistent for all sites and trees. Restricted photosynthetic production, stomatal conductance and $CO_2$ response of the leaves further suggest stress for some of the trees.

It is unclear if the emission patterns from the two sets of clones follows the climate gradient, as some trees were damaged or of different age. However, the absence of large variations in the relative BVOC emission within and across sites suggests that absence of genetic variation does suppress the variability between individuals that has been previously observed. More studies are needed to increase the understanding of genetic variation and different growing conditions on emission patterns. This study confirms the importance of taking genetic variation into account and the possible stability in emission patterns that English oak and European beech show when the genetic variability is absent. It provides an insight in how these trees, which are some of the most common tree species in Europe, might react to different climatic conditions across regions.

### Acknowledgements

We thank Ana Žust (Environmental agency of the Republic of Slovenia), Martin Piepenburg (Bavarian State Institute of Forestry) and Anders Kristian Nørgaard (University of Copenhagen) for their assistance during field campaigns and Frank M. Chmielewski (Humboldt-University of Berlin) for help and information about the IPG network. We thank Magnus Kramshøj and Michelle Schollert for performing the GC-MS analysis. Lastly, we thank Carsten Skjøth for providing species distribution data for English oak and European beech in Europe and Max van Meeningen for compiling the data into maps. The study was performed within the framework of LUCCI, which is a research centre at Lund University for studies of carbon cycles and climate interaction.

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



**Table 1. Long-term monthly average temperature (°C) and total monthly precipitation (mm) for Ljubljana in Slovenia, Grafrath in Germany and Taastrup in Denmark (averaging period is indicated).**

| Site | Weather parameter | May | June | July | Yearly | Time period | Source |
|---|---|---|---|---|---|---|---|
| Ljubljana | Temperature (°C) | 15.8 | 19.1 | 21.3 | 10.9 | 1981-2010 | Slovenian Environmental Agency: http://meteo.arso.gov.si/, last access: 10 October 2015 |
| | Precipitation (mm) | 109 | 144 | 115 | 1362 | | |
| Grafrath | Temperature (°C) | 12.9 | 16.6 | 20.9 | 8.5 | 1995-2010 | Agrarmeteorologie Bayern: http://www.wetter-by.de/, last access: 9 October 2015 |
| | Precipitation (mm) | 170.1 | 134.2 | 30.6 | 877 | | |
| Taastrup | Temperature (°C) | 11.7 | 14.8 | 17.5 | 8.7 | 1997-2010 | Danish Meteorological Institute: dmi.dk, last access: 12 October 2015 |
| | Precipitation (mm) | 52 | 76.3 | 72.3 | 687 | | |





**Table 2. The day of year (DOY) for leaf unfolding for English oak and European beech at sites Ljubljana, Grafrath and Taastrup, with both an average between 2007 and 2013 and the unfolding in 2014. No long-term observations for leaf unfolding of English oak in Taastrup are available.**

| Year | English oak | | | European beech | | |
|------|-----------|----------|----------|-----------|----------|----------|
| | Ljubljana | Grafrath | Taastrup | Ljubljana | Grafrath | Taastrup |
| 2007-2013 | 113 | 117 | - | 115 | 114 | 125 |
| 2014 | 101 | 97 | 104 | 103 | 114 | 104 |



**Table 3. Standardized BVOC emission (µg gdw$^{-1}$ h$^{-1}$), the net assimilation rate (µmol $CO_2$ m$^{-2}$ s$^{-1}$) for English oak tree clones and European beech tree clones at sites Ljubljana in Slovenia, Grafrath in Germany and Taastrup in Denmark. The values are averages from the samples collected from each tree and site ± the standard deviation (SD). The hyphen indicates that there was no compound detected for the site and tree.**

| English oak | Isoprene<br>mean ± SD | Total monoterpenes<br>mean ± SD | Total sesquiterpenes<br>mean ± SD | Total BVOC<br>mean ± SD |
|---|---|---|---|---|
| Ljubljana | 0.2 ± 0.23 | 0.05 ± 0.02 | - | 0.25 ± 0.23 |
| Grafrath 1 | 2.31 ± 0.81 | 0.22 ± 0.05 | - | 2.53 ± 0.78 |
| Grafrath 2 | 3.14 ± 1.74 | 0.2 ± 0.04 | - | 3.35 ± 1.76 |
| Taastrup 1 | 2.12 ± 0.81 | 0.17 ± 0.11 | 0.01 | 2.28 ± 0.75 |
| Taastrup 2 | 2.09 ± 1.03 | 0.06 ± 0.02 | - | 2.14 ± 1.03 |

| European beech | Isoprene<br>mean ± SD | Total monoterpenes<br>mean ± SD | Total sesquiterpenes<br>mean ± SD | Total BVOC<br>mean ± SD |
|---|---|---|---|---|
| Ljubljana | - | 0.68 ± 0.21 | - | 0.68 ± 0.21 |
| Grafrath 1 | - | 0.69 ± 0.39 | - | 0.69 ± 0.39 |
| Grafrath 2 | - | 0.59 ± 0.26 | - | 0.59 ± 0.26 |
| Taastrup 1 | - | 0.29 ± 0.06 | - | 0.29 ± 0.06 |



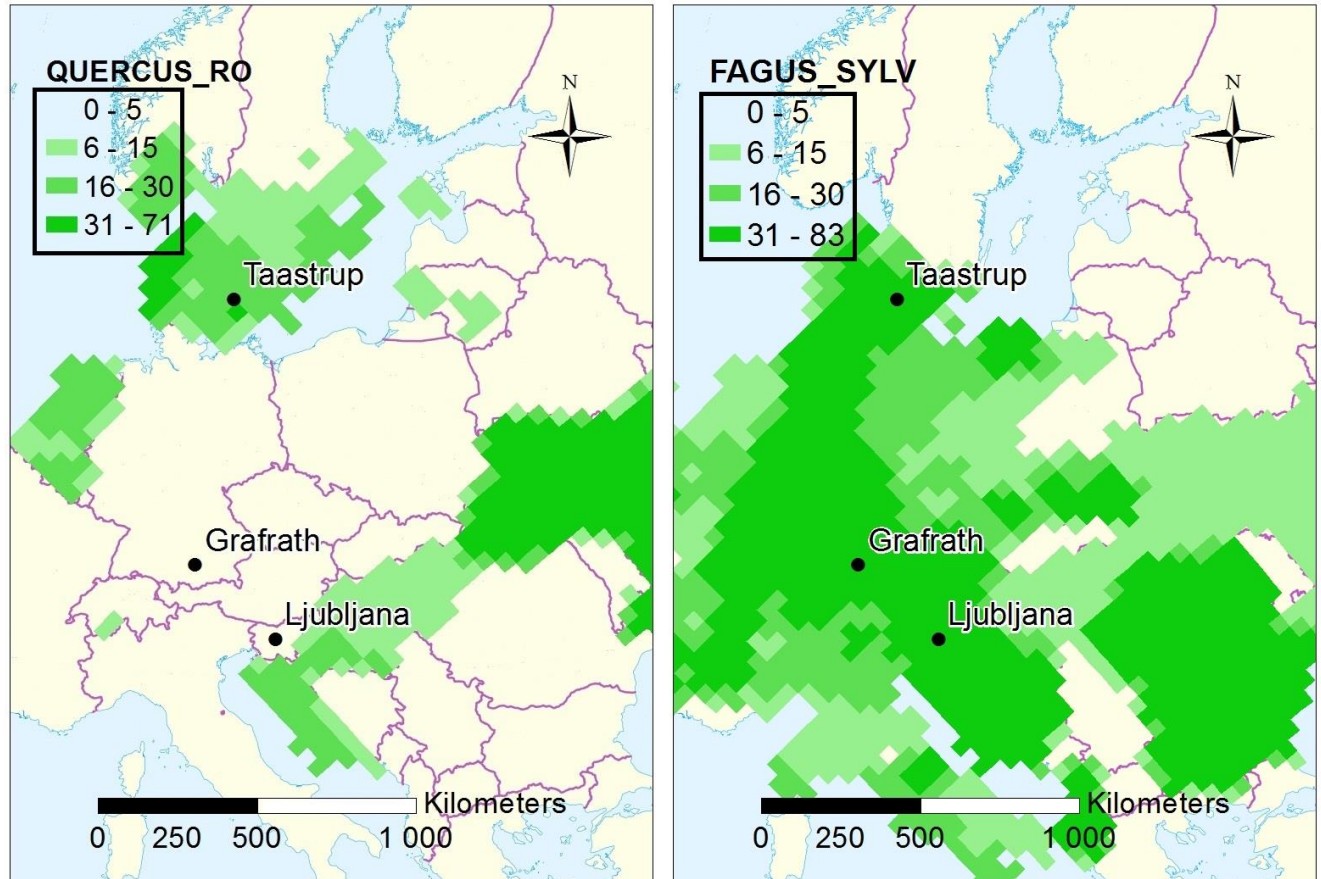

**Fig. 1.** European map over visited International Phenological Garden sites and the reported gridded tree density in percentage of English oak (left) and European beech (right). The sites used in this study are located in Slovenia (Ljubljana), Germany (Grafrath, 30 km W of Munich) and Denmark (Taastrup, 15 km W of Copenhagen). The maps are redrawn from Skjøth et al., 2008, with permission.



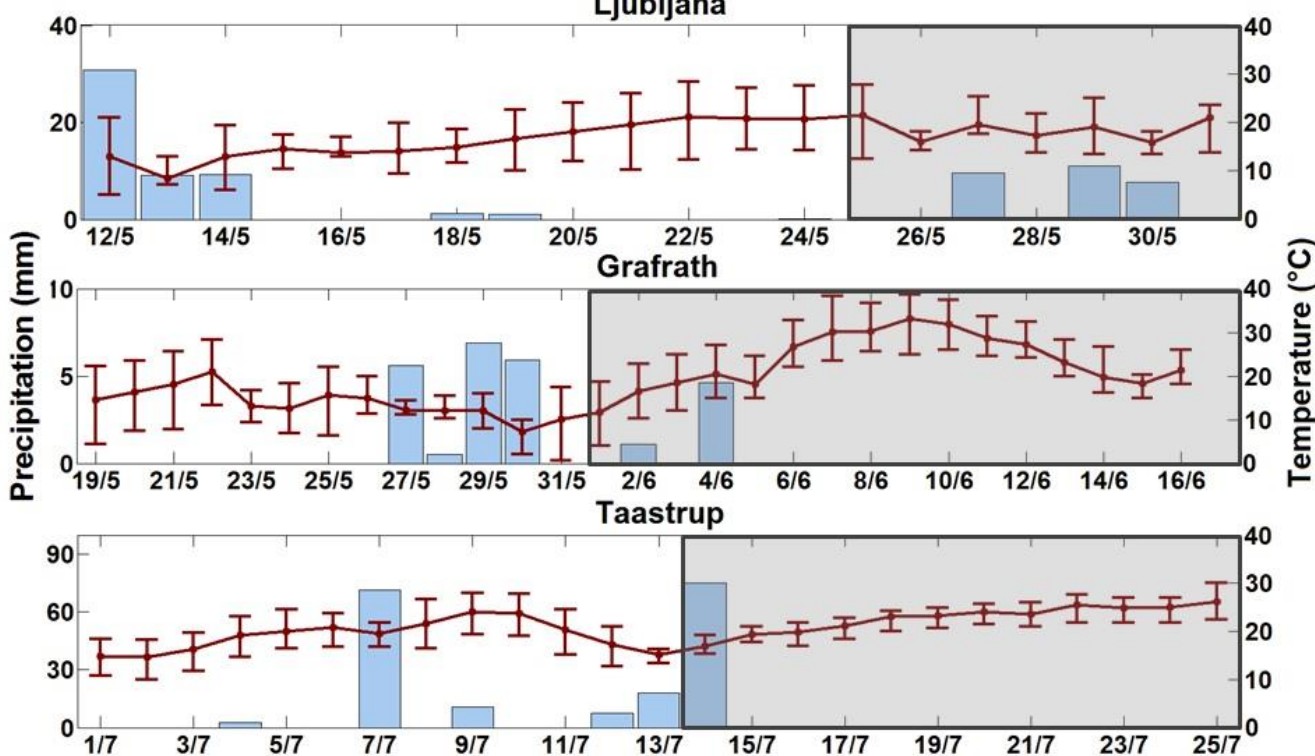

Fig. 2. The daily precipitation (mm, blue) and average daily temperature (°C, red) for IPG sites Ljubljana (Slovenia), Grafrath (Germany) and Taastrup (Denmark) two weeks prior (marked in white) and during measurement campaigns (marked in grey). The error bars mark the maximum and minimum temperatures during the day.



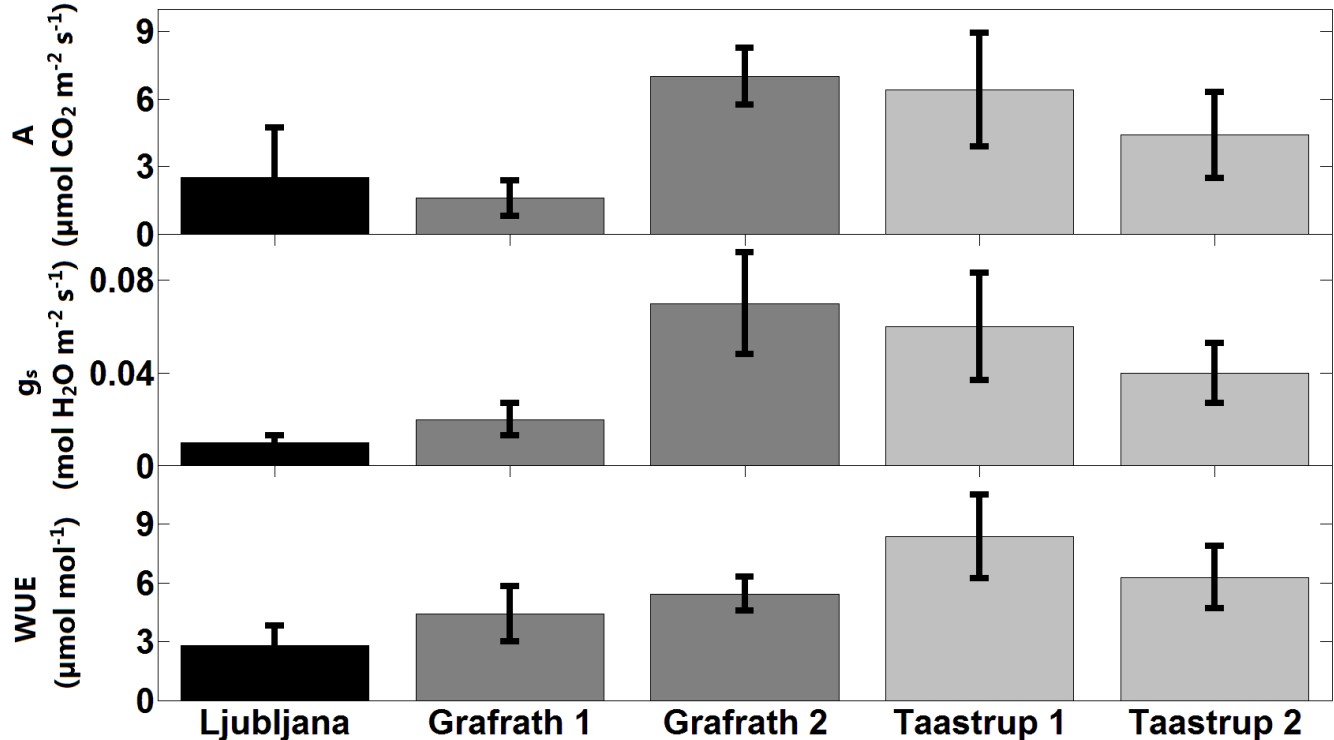

Fig. 3. Net assimilation rates (A), stomatal conductance ($g_s$) and Water Use Efficiency (WUE) for English oak trees growing at sites Ljubljana, Grafrath and Taastrup. The figure represents the average values ± standard deviation.




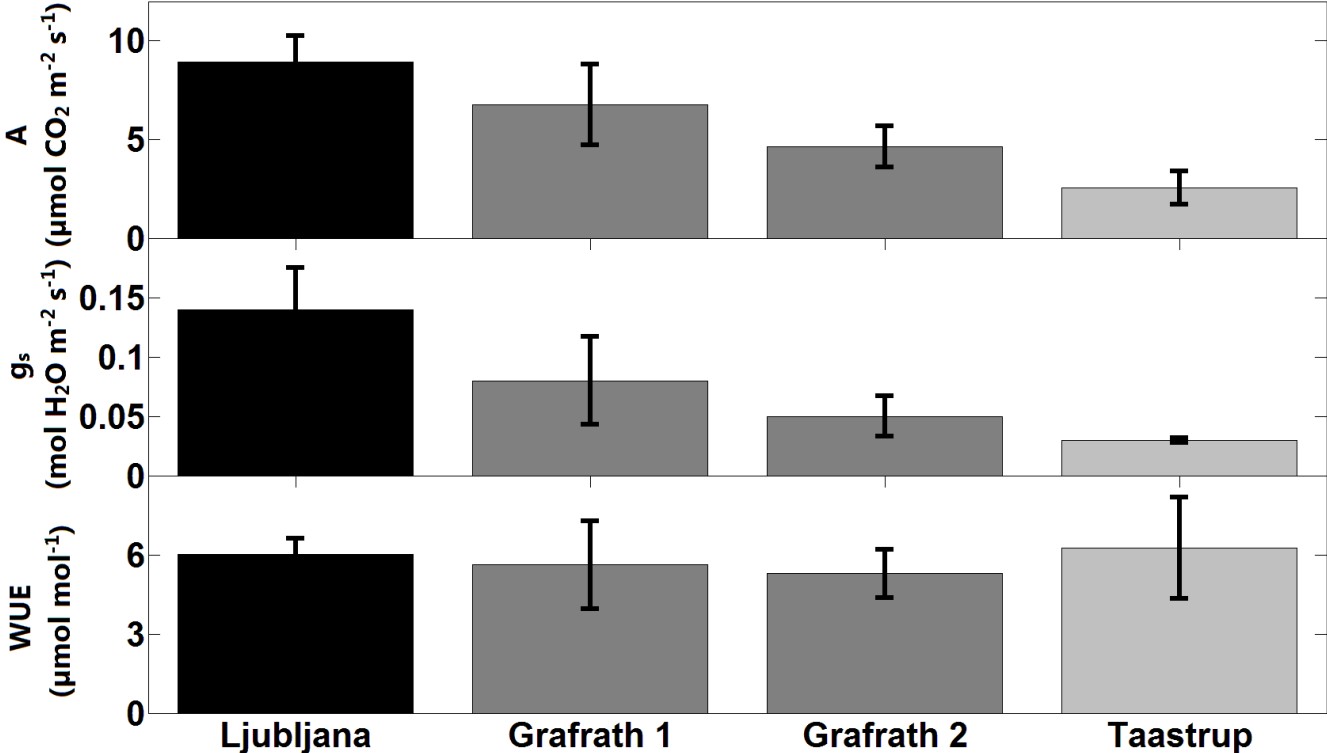

Fig. 4. Net assimilation rates (A), stomatal conductance ($g_s$) and Water Use Efficiency (WUE) for European beech trees growing at sites Ljubljana, Grafrath and Taastrup. The figure represents the average values ± standard deviation.




**Fig. 5. Net assimilation rate (A) and intercellular CO$_2$ concentration (C$_i$) response curves for English oak. The figure represents the average A-C$_i$ response curves for each tree ± standard deviation.**





**Fig. 6.** Net assimilation rate (A) and intercellular CO$_2$ concentration (C$_i$) response curves for European beech. The figure represents the average A-C$_i$ response curves for each tree ± standard deviation.



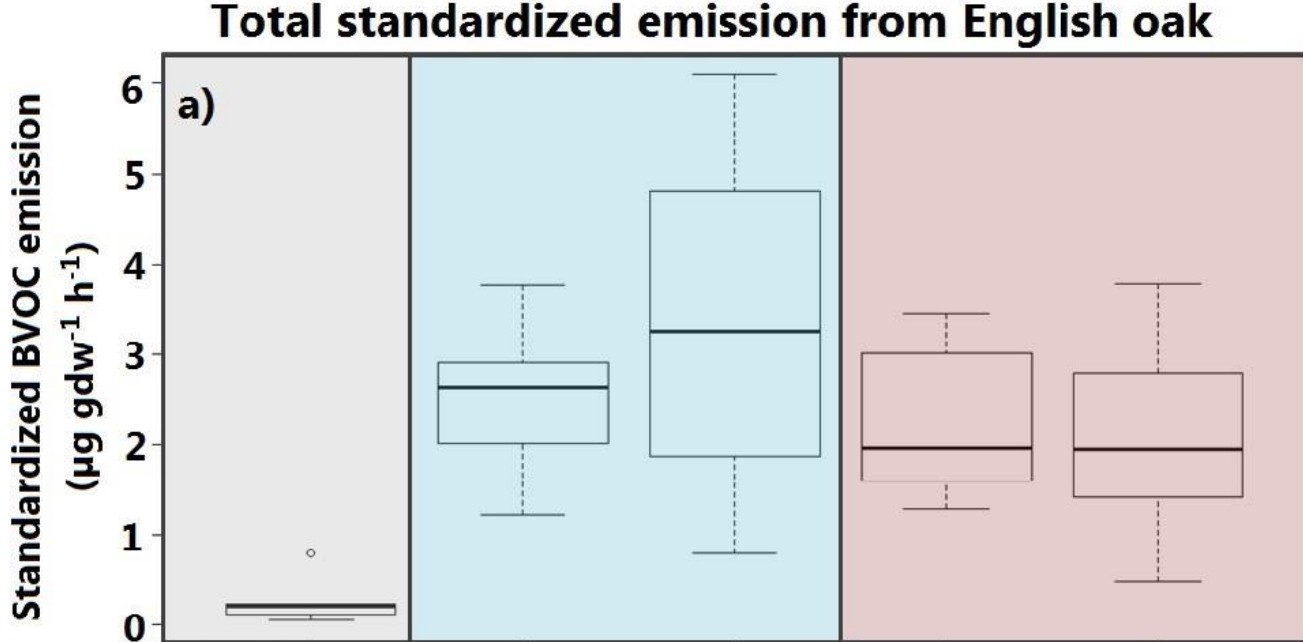

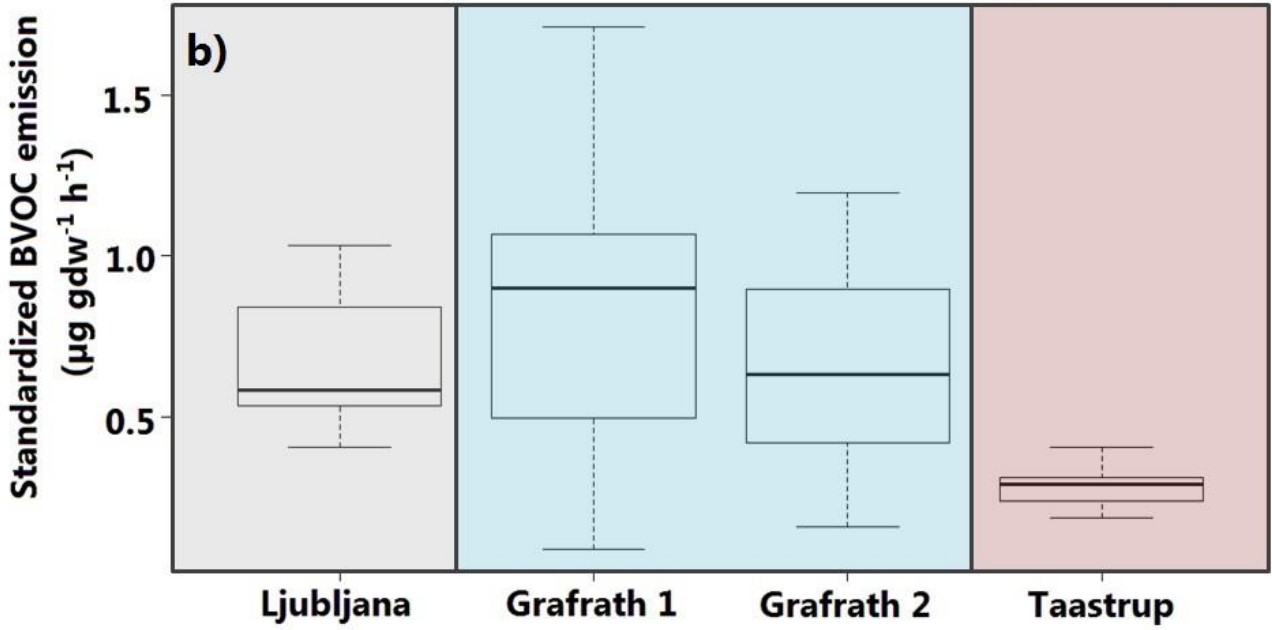

Fig. 7. The range of total standardized BVOC emission from a) English oak and b) European beech at site Ljubljana in Slovenia, Grafrath in Germany and Taastrup in Denmark. The whiskers indicate the highest and lowest values within a 1.5 interquartile range. A paired t-test (P>0.05) showed no statistically significant difference between individuals from the same site.





**Fig. 8. Relative BVOC compound contribution from English oak seen as a) sample averages for different trees and sites and b) for individual samples. The major emitted compounds are presented separately, whilst the remaining compounds are classified into other compounds.**





**Fig. 9. Relative BVOC compound contribution from European beech seen as a) sample averages for different trees and sites and b) for individual samples. The major emitted compounds are presented separately, whilst the remaining compounds are classified into other compounds.**



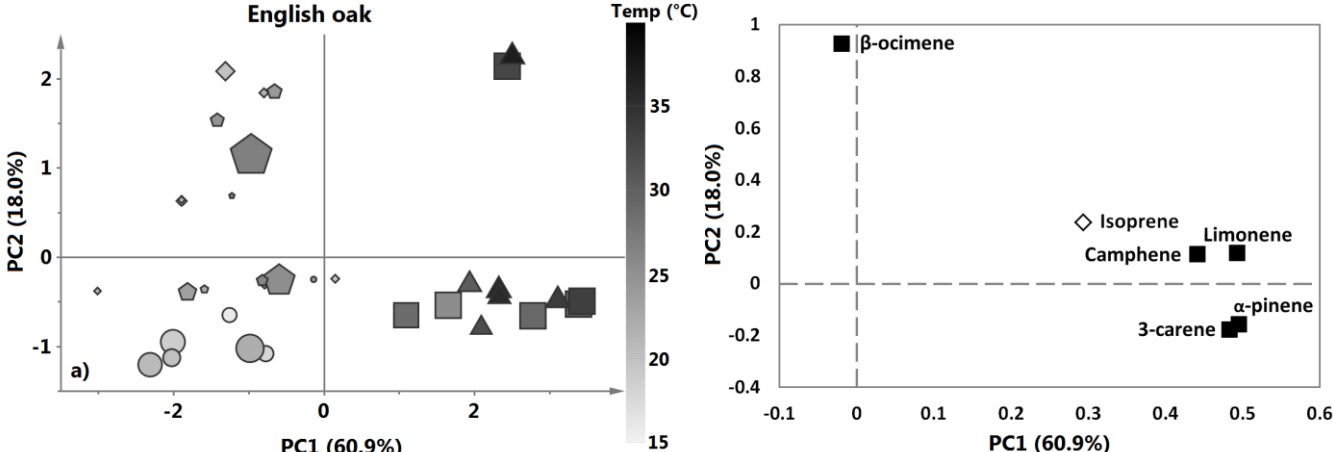

**Fig. 10. Principal component analysis on the BVOC emission from five cloned English oak trees grown at three sites. a) The scores of the principal components (PC1 and PC2), and b) the corresponding loading variables. In a), the colour explains the average air temperature. The size of the symbols depicts the average daily PAR level, with an increasing symbol size with increasing PAR levels. The symbols show samples from the different sites (circle=Ljubljana, square=Grafrath tree one, triangle=Grafrath tree two, diamond=Taastrup tree one and pentagon= Taastrup tree two). The explained variance for the different PCs is shown in parentheses.**





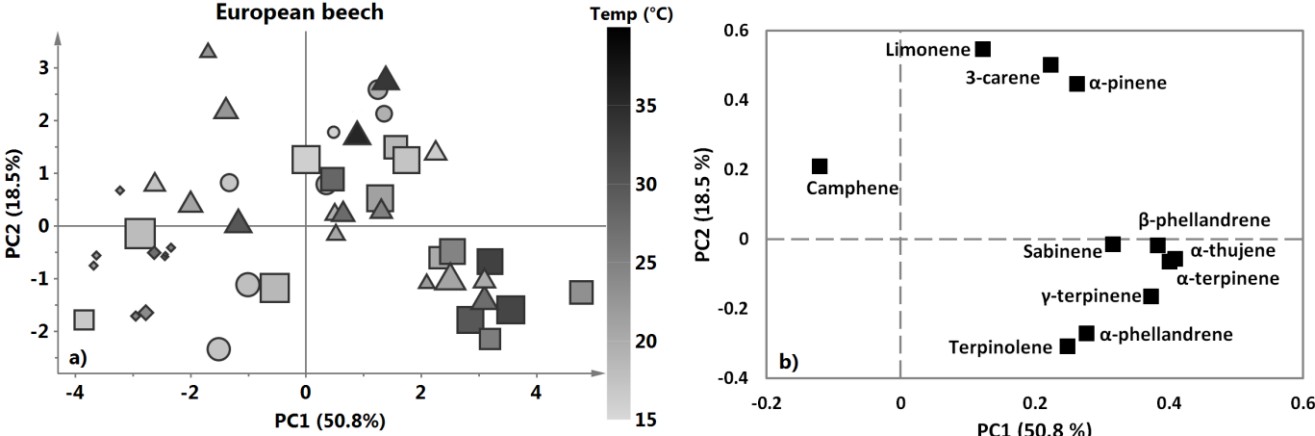

**Fig. 11. Principal component analysis on the BVOC emission from four cloned individual European beech trees growing at three sites. a) The scores of the principal components (PC1 and PC2), and b) the corresponding loading variables. In a), the colour explains the average air temperature. The size of the symbols depicts the average daily PAR level, with an increasing symbol size with increasing PAR levels. The symbols show samples from the different sites (circle=Ljubljana, square=Grafrath tree one, triangle=Grafrath tree two, diamond=Taastrup). The explained variance for the different PCs is shown in parentheses.**