# Peer review of "BVOC emissions from English oak (*Quercus robur*) and European beech (*Fagus sylvatica*) along a latitudinal gradient"

_Biogeosciences, 2016_

## Referee Comment (RC1) · Anonymous Referee #1 · 17 May 2016

The paper from Person et al has been shown the emission of BVOC from two different tree species. The authors underline that BVOC emission spectra remain somehow constant along quite a high gradient. This conclusion is quite important for modeling studies as usually for small gradients variation in emission is not introduced in the models. Anyway, overall is a very nice paper which fit in the scope of the journal. There are few corrections which could be done: I have a concern regarding to the very limited number of trees available. - p. 2 L25: paragraph Isoprene has..... should be re-written Materials and methods How the daily PAR had been calculated? This is very important. Why the assimilation rates are low? I could guess this is due to the fact that only leaves from lowest branches. It has been shown that assimilation rate and BVOC

emission scale with the heights (see Niinemets et al. JOURNAL OF GEOPHYSICAL RESEARCH, VOL. 115, G04029, 2010). Other problem is the stress induce on the leaves which could increase BVOC emission.

---

## Referee Comment (RC2) · Anonymous Referee #2 · 9 Aug 2016

The paper by Ylva Persson et al focus on BVOC emissions from clones of two different tree species located in the international phenological gardens. This experimental design is good as it removes some of the uncertainties in relation to this kind of studies: the effect of natural variation among species due to genetic variation. This allows the authors to focus on the effect of climate and meteorology.

The topic is indeed relevant for Biogeosciences by contribution with new observations of BVOCs. The study design is systematic, easy to reproduce and the conclusions are clear. The main weakness is that the study is based on a relatively small observational data set.

The text is well written and the results are discussed in a balanced way with sufficient

credit the recent relevant work.

I have only minor comments or suggestions to this study as seen below:

The authors compare BVOC emissions from populations of same genetic structure but at different climatic locations. Generally speaking, then comparisons of such data set would often include a test of significance. Would a test for significance be appropriate in this particular case?

Fig 1 is a bit difficult to read. It would be better if the country borders were drawn on top of the coloured grids

Could some of the figures be more efficiently presented in a table. Figures like Fig 2,3,7 appear to be highly related to Table 3. If this is possible, then it would make the results more quantitative and at the same time save space.

It is unclear to me, why the BVOC component Sabiene is not measured at some of the trees in Taastrup and Grafrath 1 (e.g. Fig 9). Is there a particular reason to this. Secondly, is findings in relation to the BVOC component Sabiene an important finding that suggest that large variations are found at the individual tree level? Thus suggesting that BVOCs from several trees must be measured before conclusions can be drawn?

---

## Author Comment (AC1) · 5 Sep 2016

Ylva van Meeningen, Guy Schurgers, Riikka Rinnan, Thomas Holst

2016

We thank the editor and the reviewers for their ideas and suggestions to improve this paper. All of these have been carefully considered in order to improve the readability of this manuscript. Below follows a list of changes made according to each referee's suggestions.

Reviewer #1 We thank the reviewer for the thorough review of our manuscript and for the following suggested points of improvement.

[Figure]

Point 1: I have a concern regarding to the very limited number of trees available.

We absolutely agree that the limited number of trees available is of concern, as it makes it difficult to statistically verify the results. Unfortunately, the IPG network only provides with two individuals per species and site. For this study, measurements were done on all trees that were available. Furthermore, it should be pointed out that they were done on adult and naturally growing trees instead of saplings grown outdoors or in greenhouses. In order to make the text clearer regarding the limited number of trees, an additional sentence in section 2.1 (p. 3, L25) has been added. The new sentence is marked in italics: "The IPG network was initiated in 1957 and performs long-term phenological observations on some of the most common European plant species across Europe. *Each site is initially provided with up to two individuals per species.*"

Point 2: p. 2 L25: paragraph Isoprene has... should be re-written.

Thank you for your comment. We agree that the paragraph could be rewritten in order to improve the flow of the text. Our suggestion is to remove the sentence all together and add additional text to the previous line. The text (marked in italics) would then be the following: "Both trees are reported to be de novo emitters, *lacking specialized anatomical structures for storing newly produced BVOCs* (Holzke et al., 2006; Kleist et al., 2012; Steinbrecher et al., 2013).

Point 3: Materials and methods How the daily PAR had been calculated? This is very important.

Thank you for your comment! To make things a bit clearer, we did not use daily PAR in our measurements, but the PAR level was fixed within the chamber to 1000 $\mu$mol m-2 s-1 for all of the measured samples. The chamber is equipped with a 6400-02B LED light source which is software adjustable from 0 to 2000 $\mu$mol m-2 s-1. For the daily PAR which is mentioned on p. 8, L18, we used an external PAR sensor (LI-190SA) connected tothe same unit as the internal light source. An additional sentence was

addedto section 2.2 (p. 4, L29): "In addition to BVOC measurements, net assimilation rates (A) and stomatal conductance (gs) were determined for each leaf using a portable photosynthesis system (LI-6400, LICOR, Lincoln, NE, USA), equipped with a led source leaf chamber (6400-02B) and an external quantum sensor (LI-190SA)."

Point 4: Why the assimilation rates are low? I could guess this is due to the fact that only leaves from lowest branches. It has been shown that assimilation rate and BVOC emission scale with the heights (see Niinemets et al. JOURNAL OF GEOPHYSICAL RESEARCH, VOL. 115, G04029, 2010).

As the reviewer is pointing out, one of the reasons of low assimilation rates could be due to that measurements were only performed on the lowest positioned branches. Looking through some literature on similar PAR levels, English oak has been reported to have assimilation rates between 5-16 $\mu$mol $CO_2$ m-2 s-1 for sunlit branches and 0-5 $\mu$mol $CO_2$ m-2 s-1 for shaded branches (Morecroft and Roberts, 1999; Vallandres et al., 2002) and European beech between 5-6.5 $\mu$mol $CO_2$ m-2 s-1 for sunlit branches and 0.5-4 $\mu$mol $CO_2$ m-2 s-1 for shaded branches (Warren et al., 2007). Our study has assimilation rates between 2.6-7 $\mu$mol $CO_2$ m-2 s-1 for oak and 2.6-8.9 $\mu$mol $CO_2$ m-2 s-1 for beech. This would suggest that the assimilation rates for some trees in this study might be adapted to shade conditions, which would have an effect on the BVOC emissions as well. We already acknowledge in the discussion (p. 11, L5-10) that we know there is an effect of height, in particular for beech. A suggestion could be to add in an extra sentence together with references regarding the low assimilation rates for both oak and beech based on the discussion held here. The sentence for oak (marked in italics) was added into section 4.1 (p. 9, L20-25): "A low net assimilation rate of 2.5 $\mu$mol $CO_2$ m-2 s-1 also indicated stress, which was less than half the rates found at the other sites, apart from one of the Grafrath trees. The assimilation rates of the tree in Ljubljana and Grafrath 1 correspond to the assimilation rates reported from shade adapted leaves, whilst the remaining rates are in range of sun adapted leaves (Morecroft and Roberts, 1999; Vallandres et al., 2002)." Morecroft, M.D. and Roberts,

J. M.: Photosynthesis and Stomatal Conductance of Mature Canopy Oak (Quercus robur) and Sycamore (Acer pseudoplatanus) Trees Throughout the Growing Season, Funct. Ecol., 13, 332–342, 1999. Valladares, F., Manuel, J., Aranda, I., Balaguer, L. and Dizengremel, P.: The greater seedling high-light tolerance of Quercus robur over Fagus sylvatica is linked to a greater physiological plasticity, Trees, 16, 395–403, 2002.

The sentence for beech was added to section 4.1 (p. 10, L5-10): "In this study, the average net assimilation rates for Ljubljana and Grafrath were between 4.62-8.93 $\mu$mol CO2 m-2 s-1, whilst Taastrup had the lowest rate of 2.56 $\mu$mol CO2 m-2 s-1. The values for Ljubljana and Grafrath are in the same range as assimilation rate for sunlit leaves measured earlier, whilst values from Taastrup are more similar to the assimilation rate for shaded leaves (Warren et al., 2007)." Warren, C. R., Matyssek, R. and Tausz, M.: Internal conductance to CO2 transfer of adult Fagus sylvatica: Variation between sun and shade leaves and due to free-air ozone fumigation, Environ. Exp. Bot., 59, 130–138, 2007.

Point 5: Other problem is the stress induce on the leaves which could increase BVOC emission.

Thank you for pointing this out. The reviewer is absolutely right regarding the risk of stress induced on the leaves as they are handled and measured, leading to a potential increase in BVOC emissions. We cannot deny that as we are inserting the leaf into the chamber and providing it with climatic conditions which are usually a little bit different from what the ambient conditions are at that time, that we do influence the emission patterns from that particular leaf. We have however tested the stress induce beforehand with the help of a PTR-MS. We found out that one hour of acclimation to the chamber conditions was sufficient for the emissions to return to a stable level.

Reviewer #2 We thank the reviewer for the thorough review of our manuscript and for the following suggested improvements. The following describes the view we have on the four points raised.

Point 1: The authors compare BVOC emissions from populations of same genetic structure but at different climatic locations. Generally speaking, then comparisons of such data set would often include a test of significance. Would a test of significance be appropriate in this particular case? Thank you for the comment! We have performed significance tests on the data, both between sites using a one-way ANOVA followed by a Tukey's test, and within sites where two clones were available using a paired t-test. However, we have to acknowledge that the wrong t-test has been performed. Instead of a paired t-test, which studies the significance from dependent samples, a two-sample t-test, studying the significance of independent samples, should have been used. The results of the different significance tests are however not contradicting the similarities between clones from the same site ($P>0.05$), but we will in the revised manuscript correct the paired t-test to a two-sample t-test (p. 6, L15 and figure caption for Fig. 7).

Point 2: Fig. 1 is a bit difficult to read. It would be better if the country borders were drawn on top of the coloured grids.

Thank you for the comment. We will redraw the map, adding in country borders for the revised manuscript.

Point 3: Could some of the figures be more efficiently presented in a table. Figures like Fig 2,3,7 appear to be highly related to Table 3. If this is possible, then would it make the results more quantitative and at the same time save space.

We agree with the reviewer that the figures could be represented in a table instead. Fig. 7 for example showing the boxplots of the standardized emission patterns of oak and beech at the different sites are already covered in Table 3. We assume that the reviewer is referring to Figs 3 and 4 regarding assimilation rates, stomatal conductance and WUE for oak and beech and these could be added into the table as well. We therefore suggest that Figs 3, 4 and 7 and the text referring to these figures is removed from the manuscript and that Table 3 is rewritten, adding in stomatal conductance and WUE to the table.
Point 4: It is unclear to me, why the BVOC component Sabinene is not measured at some of the trees in Taastrup and Grafrath 1 (e.g. Fig 9). Is there a particular reason to this. Secondly, is findings in relation to the BVOC component Sabinene an important finding that suggest that large variations are found at the individual tree level? Thus suggesting that BVOCs from several trees must be measured before conclusions can be drawn?

As the reviewer correctly points out, the component sabinene was not detected in all the samples of the beech trees in Taastrup, Grafrath 1 and one sample in Grafrath 2. By inspecting the data, we have seen that the samples without Sabinene for Grafrath 1 and 2 were all taken in the morning before 9:00. The samples without Sabinene for Taastrup were from measurements performed on leaves that had started to turn yellow. From the data that we present here, we can add in two additional sentences (marked in italics) at p. 8, L5: "In the few samples where sabinene was not detected, limonene was the main emitted compound. In Grafrath, these samples were taken usually before 9:00 in the morning. For the tree in Taastrup, the samples without sabinene were from leaves which were more yellow in colour than the other leaves."

On behalf of all authors,

Ylva van Meeningen

---

## Author Response (AR1)

**Final author response to "BVOC emissions from English oak (*Quercus robur*) and European beech (*Fagus sylvatica*) along a latitudinal gradient"**

**Ylva van Meeningen, Guy Schurgers, Riikka Rinnan, Thomas Holst**

**September 28, 2016**

We thank the editor and the reviewers for their ideas and suggestions to improve the manuscript. All of these have been carefully considered in order to improve the readability of the paper. Below follows a list of changes made according to each referee's suggestions.

**Reviewer #1**

We thank the reviewer for the thorough review of our manuscript and for the following suggested points of improvement.

Point 1: *I have a concern regarding to the very limited number of trees available.*

We absolutely agree that the limited number of trees available is of concern, as it makes it difficult to statistically verify the results. Unfortunately, the IPG network only provides with two individuals per species and site. For this study, measurements were done on all trees that were available. Furthermore, it should be pointed out that they were done on adult and naturally growing trees instead of saplings grown outdoors or in greenhouses. In order to make the text clearer regarding the limited number of trees, an additional sentence in section 2.1 (p. 3, L25) has been added. The new sentence is marked in italics:

"The IPG network was initiated in 1957 and performs long-term phenological observations on some of the most common European plant species across Europe. *Each site is initially provided with up to two individuals per species.*"

Point 2: *p. 2 L25: paragraph Isoprene has… should be re-written.*

Thank you for your comment. We agree that the paragraph could be rewritten in order to improve the flow of the text. We have removed the sentence all together and added additional text to the previous line. The text (marked in italics) now reads:

"Both trees are reported to be *de novo* emitters, *lacking specialized anatomical structures for storing newly produced BVOCs* (Holzke et al., 2006; Kleist et al., 2012; Steinbrecher et al., 2013)."

Point 3: *Materials and methods How the daily PAR had been calculated? This is very important.*

Thank you for your comment! To make things a bit clearer, we did not use daily PAR in our measurements, but the PAR level was fixed within the chamber to 1000 µmol $m^{-2}$ $s^{-1}$ for all of the measured samples. The chamber is equipped with a 6400-02B LED light source which is software adjustable from 0 to 2000 µmol $m^{-2}$ $s^{-1}$. For the daily PAR which is mentioned on p. 8, L18, we used an external PAR sensor (LI-190SA) connected to the same unit as the internal light source. An additional sentence was added to section 2.2 (p. 4, L29):

"In addition to BVOC measurements, net assimilation rates (A) and stomatal conductance ($g_s$) were determined for each leaf using a portable photosynthesis system (LI-6400, LICOR, Lincoln, NE, USA), equipped with a led source leaf chamber (6400-02B) *and an external quantum sensor (LI-190SA) which measured daily Photosynthetic Active Radiation (PAR) rates.*"

Point 4: *Why the assimilation rates are low? I could guess this is due to the fact that only leaves from lowest branches. It has been shown that assimilation rate and BVOC emission scale with the heights (see Niinemets et al. JOURNAL OF GEOPHYSICAL RESEARCH, VOL. 115, G04029, 2010).*

As the reviewer is pointing out, one of the reasons of low assimilation rates could be due to that measurements were only performed on the lowest positioned branches. Looking through some literature on similar PAR levels, English oak has been reported to have assimilation rates between 5-16 µmol $CO_2$ $m^{-2}$ $s^{-1}$ for sunlit branches and 0-5 µmol $CO_2$ $m^{-2}$ $s^{-1}$ for shaded branches (Morecroft and Roberts, 1999; Vallandres et al., 2002) and European beech between 5-6.5 µmol $CO_2$ $m^{-2}$ $s^{-1}$ for sunlit branches and 0.5-4 µmol $CO_2$ $m^{-2}$ $s^{-1}$ for shaded branches (Warren et al., 2007). Our study has assimilation rates between 2.6-7 µmol $CO_2$ $m^{-2}$ $s^{-1}$ for oak and 2.6-8.9 µmol $CO_2$ $m^{-2}$ $s^{-1}$ for beech. This would suggest that the assimilation rates for some trees in this study might be adapted to shade conditions, which would have an effect on the BVOC emissions as well. We already acknowledge in the discussion (p. 11, L5-10) that we know there is an effect of height, in particular for beech.

To clarify, we added an extra sentence together with references regarding the low assimilation rates for both oak and beech based on the discussion held here. The sentence for oak (marked in italics) was added into section 4.1 (p. 9, L20-25):

"A low net assimilation rate of 2.5 µmol $CO_2$ $m^{-2}$ $s^{-1}$ also indicated stress, which was less than half the rates found at the other sites, apart from one of the Grafrath trees. *The assimilation rates of the tree in Ljubljana and Grafrath 1 correspond to the assimilation rates reported from shade adapted leaves, whilst the remaining rates are in range of sun adapted leaves (Morecroft and Roberts, 1999; Vallandres et al., 2002).*"

Morecroft, M.D. and Roberts, J. M.: Photosynthesis and Stomatal Conductance of Mature Canopy Oak (Quercus robur) and Sycamore (Acer pseudoplatanus) Trees Throughout the Growing Season, Funct. Ecol., 13, 332–342, 1999.

Valladares, F., Manuel, J., Aranda, I., Balaguer, L. and Dizengremel, P.: The greater seedling high-light tolerance of *Quercus robur* over *Fagus sylvatica* is linked to a greater physiological plasticity, Trees, 16, 395–403, 2002.

The sentence for beech was added to section 4.1 (p. 10, L5-10):

"In this study, the average net assimilation rates for Ljubljana and Grafrath were between 4.62-8.93 µmol $CO_2$ $m^{-2}$ $s^{-1}$, whilst Taastrup had the lowest rate of 2.56 µmol $CO_2$ $m^{-2}$ $s^{-1}$. *The values for Ljubljana and Grafrath are in the same range as assimilation rate for sunlit leaves measured earlier, whilst values from Taastrup are more similar to the assimilation rate for shaded leaves (Warren et al., 2007).*"

Warren, C. R., Matyssek, R. and Tausz, M.: Internal conductance to $CO_2$ transfer of adult *Fagus sylvatica*: Variation between sun and shade leaves and due to free-air ozone fumigation, Environ. Exp. Bot., 59, 130–138, 2007.

Point 5: *Other problem is the stress induce on the leaves which could increase BVOC emission.*

Thank you for pointing this out. The reviewer is absolutely right regarding the risk of stress induced on the leaves as they are handled and measured, leading to a potential increase in BVOC emissions. We cannot deny that as we are inserting the leaf into the chamber and providing it with climatic conditions which are usually a little bit different from what the ambient conditions are at that time, that we do influence the emission patterns from that particular leaf. We have however tested the stress induce beforehand with the help of a PTR-MS. We found out that one hour of acclimation to the chamber conditions was sufficient for the emissions to return to a stable level.

**Reviewer #2**

We thank the reviewer for the thorough review of our manuscript and for the following suggested improvements. The following describes the view we have on the four points raised.

Point 1: *The authors compare BVOC emissions from populations of same genetic structure but at different climatic locations. Generally speaking, then comparisons of such data set would often include a test of significance. Would a test of significance be appropriate in this particular case?*

Thank you for the comment! We have performed significance tests on the data, both between sites using a one-way ANOVA followed by a Tukey's test, and within sites where two clones were available using a paired t-test.

However, we have to acknowledge that the wrong t-test has been performed. Instead of a paired t-test, which studies the significance from dependent samples, a two-sample t-test, studying the significance of independent samples, should have been used. The results of the different significance tests are however not contradicting the similarities between clones from the same

site (P>0.05), but we will in the revised manuscript correct the paired t-test to a two-sample t-test (p. 6, L15 and figure caption for Fig. 7).

5 Point 2: *Fig. 1 is a bit difficult to read. It would be better if the country borders were drawn on top of the coloured grids.*
Thank you for the comment. We have redrawn the map, adding in country borders for this revised manuscript.

Point 3: *Could some of the figures be more efficiently presented in a table. Figures like Fig 2,3,7 appear to be highly related to Table 3. If this is possible, then would it make the results more quantitative and at the same time save space.*

10 We agree with the reviewer that the figures could be represented in a table instead. Fig. 7 for example showing the boxplots of the standardized emission patterns of oak and beech at the different sites are already covered in Table 3. We assume that the reviewer is referring to Figs 3 and 4 regarding assimilation rates, stomatal conductance and WUE for oak and beech and these had now been added into the table as well. We therefore have removed Figs 3, 4 and 7 and the text referring to these figures from the manuscript and Table 3 had been rewritten, adding in stomatal conductance and WUE to the table.

Point 4: *It is unclear to me, why the BVOC component Sabinene is not measured at some of the trees in Taastrup and Grafrath 1 (e.g. Fig 9). Is there a particular reason to this. Secondly, findings in relation to the BVOC component Sabinene an important finding that suggest that large variations are found at the individual tree level? Thus suggesting that BVOCs from several trees must be measured before conclusions can be drawn?*

20 As the reviewer correctly points out, the component sabinene was not detected in all the samples of the beech trees in Taastrup, Grafrath 1 and one sample in Grafrath 2. By inspecting the data, we have seen that the samples without Sabinene for Grafrath 1 and 2 were all taken in the morning before 9:00. The samples without Sabinene for Taastrup were from measurements performed on leaves that had started to turn yellow. From the data that we present here, we can add in two additional sentences (marked in italics) at p. 8, L5:

25 "In the few samples where sabinene was not detected, limonene was the main emitted compound. *In Grafrath, these samples were taken usually before 9:00 in the morning. For the tree in Taastrup, the samples without sabinene were from leaves which were more yellow in colour than the other leaves.*"

During the careful revision of this manuscript, we have also recognized a wrong factor in the calculus for the emission rates
30 which has been checked and corrected now. While the absolute emission rates had changed somehow, this did not affect comparisons between plants or sites, and the overall results and conclusions remain the same and have not been altered by this mistake.
On behalf of all authors,
Ylva van Meeningen

[revised manuscript text omitted]